# A parsimonious model of blood glucose homeostasis

Eric Ng[1,2]*, Jaycee M. Kaufman[1,2], Lennaert van Veen[1], Yan Fossat[2]

**1** Ontario Tech University, Oshawa, Ontario, Canada, **2** Klick Applied Sciences, Toronto, Ontario, Canada

* eric.ng@ontariotechu.ca

## Abstract

The mathematical modelling of biological systems has historically followed one of two approaches: comprehensive and minimal. In comprehensive models, the involved biological pathways are modelled independently, then brought together as an ensemble of equations that represents the system being studied, most often in the form of a large system of coupled differential equations. This approach often contains a very large number of tuneable parameters (> 100) where each describes some physical or biochemical subproperty. As a result, such models scale very poorly when assimilation of real world data is needed. Furthermore, condensing model results into simple indicators is challenging, an important difficulty in scenarios where medical diagnosis is required. In this paper, we develop a minimal model of glucose homeostasis with the potential to yield diagnostics for pre-diabetes. We model glucose homeostasis as a closed control system containing a self-feedback mechanism that describes the collective effects of the physiological components involved. The model is analyzed as a planar dynamical system, then tested and verified using data collected with continuous glucose monitors (CGMs) from healthy individuals in four separate studies. We show that, although the model has only a small number (3) of tunable parameters, their distributions are consistent across subjects and studies both for hyperglycemic and for hypoglycemic episodes.

**Data Availability Statement:** All blood glucose data used in this study are publicly available. Data in the Klick Pilot Study and Klick Follow-up Studies can be accessed through sections S1 and S2 Data respectively. The data of the Stanford study can be

## Author summary

We present a model of glucose homeostasis that consists of an equation of the mass-action-kinetics type for glucose levels and a closed propotional-integral control loop. The control loop models the aggregate effect of all physiological components of homeostasis, such as the production and uptake of insulin and glucagon. We study the model as a smooth, planar, dynamical system and show that its solutions are bounded for parameter values that correspond to healthy individuals. We then fit the model parameters to datasets obtained with continuous glucose monitors in four different studies, and show that they have structured distributions for individuals across a healthy population.

obtained via the following URL: https://doi.org/10.1371/journal.pbio.2005143.s010.

**Funding:** This research is jointly funded by Klick Applied Sciences and Mitacs Accelerate program (IT17808) for EN and JMK as a collaboration between Ontario Tech University and Klick Applied Sciences. Mitacs had no role in study design, data collection and analysis, decision to publish, or preparation of the manuscript.

**Competing interests:** Lennaert van Veen and Yan Fossat are co-inventors of the patent "System and Method for Evaluating Glucose Homeostasis" used in part for this research. The patent is filed by Klick Inc. to Patent Cooperation Treaty (PCT) pending application approval (PCT/CA2020/051497).

## Introduction

In the 1920s, Walter Cannon condensed a plethora of observations and ideas of early physiologists into the concept of *homeostasis*, i.e. the automated regulation of conditions of the human body, such as temperature, blood pressure and heart rate [1, 2]. This regulation serves to counteract disturbances from outside the body and keep its internal conditions in their safe operating range. While a monumental leap forwards philosophically, the concept remained largely qualitative in nature. Cannon carefully based his reasoning on observational evidence, but the latter is often anecdotal or derived from very small-scale experiments with Cannon himself or members of his lab as subjects. Nonetheless, his hypotheses on homeostasis were tantalizingly close to a quantitative description. For instance, he posed that "If a state remains steady it does so because any tendency towards change is automatically met by increased effectiveness of the factor or factors which resist the change"—a notion begging to be translated into a mathematical model. It was not until the 1940s, however, that a connection was made to the theory of feedback control. Particularly prolific in making this connection was Norbert Wiener, who devoted multiple sections of his book on Cybernetics to feedback control in human physiology [3]. From similarities between physiological and servo-mechanical feedback control, Wiener derived several interesting ideas on the mechanism of homeostasis and how its breakdown is associated with pathological states such as involuntary tremors. Over the two decades after the publication of Wiener's book, the study of homeostasis became more firmly grounded in mathematics and experiment. Examples include the work of Corson *et al.* [4] on body water regulation, that of Stolwijk and Hardy [5] on body temperature and that of Powell [6] on plasma calcium regulation. Probably the most active topic of research, however, was the regulation of the blood glucose concentration. The malfunction of this particular feedback subsystem is associated with diabetes. The incidence of type-2 diabetes in the USA doubled in the 1960s [7] and this rapid rise likely explains the research focus of many physiologists, physicists and biochemical engineers entering the new field of homeostasis modelling. The quest to gain a better understanding of glucose homeostasis is ever more urgent, as diabetes now affects more than 30 million people in the USA alone (over 9% of the population), while another 84 million Americans have prediabetes [7, 8].

The steady progress in glucose homeostasis modelling, from reasoning by analogy to quantitative, predictive theory, can be explained in terms of major advancements in three directions. First is the gathering of data. Experiments were designed to unveil the interplay of glucose and regulatory hormones like insulin and glucagon, both in humans and in other mammals. An example of early work is that by Metz [9], who measured the production of insulin in response to artificially increased or reduced glucose levels in dogs. A few years later, Burns *et al.* [10] was one of several groups to design a continuous blood glucose monitor that could be used for a number of hours on a human subject in clinical setting. They used the data to model the oral glucose challenge, in which a subject drinks a calibrated amount of glucose after fasting, and the recovery of normal blood glucose levels is monitored.

Second is the formulation of mathematical models. Informed by the data that were becoming available, various models, both of the innate glucose dynamics and of the feedback through hormones, were formulated and tested. Initially, mostly linear models were considered, for instance by Bolie [11], who used the data of Metz [9] to estimate parameters such as the rate constants of glucose removal and insulin production. Similar results were obtained by Ackeman *et al.* [12], who focused on the oral glucose challenge. Apparently frustrated by the sparsity of available data, they revisited the parameter fitting a few years later, finding good agreement with continuous glucose measurements as well [13]. Bergman *et al.* [14] systematically compared several linear and nonlinear models, the nonlinearity appearing in the

interaction of insulin and glucose. They concluded that the action of insulin on elevated glucose levels is best described by a product of the respective concentrations, akin to mass-action kinetics. This feedback mechanism is now widely used in glucose control models, including ours presented in this paper.

Third is the availability of digital computers. While, in the early 1960s, Bolie [11] built his own analogue computer especially for the purpose of producing numerical predictions with his model, aptly called the "Hormone Computer", a decade later digital computers were rapidly becoming more accessible and useful, as demonstrated, for instance, by the work of Ceresa *et al.* [15] and Gatewood *et al.* [13]. Even by the standards of 1970s digital computing, time-stepping glucose homeostasis models in the form of small systems of ordinary differential equations was a light task. The difficulty lay in the computational cost of systematically exploring parameter space to find physiologically sound behaviour—a process called conformation by Gatewood *et al.* [13] that would today be considered a form of data assimilation. Indeed, the number of simulations one needs to run to gain a comprehensive overview of model behaviours grows exponentially with the number of parameters to be tuned, such as rate constants and baseline glucose and hormone concentrations. For a model with two variables and four parameters, like Bergman's preferred minimal model [14], we can estimate that conformation of a single peak in glucose concentration must have cost at least one hour on the equipment available at the time, like the IBM System/360 Model 50 used by Ličko and Silvers [16].

In summary, by the 1980s various models glucose homeostasis had been developed and tuned to measured data with the aid of computational algorithms. As a result, much insight was gained in the hormonal regulation of blood glucose and quantitative estimates of, for instance, insulin sensitivity [14] and disappearance rate constants of glucose and insulin [16] were obtained. A more ambitious goal, formulated already by Gatewood *et al.* [13] is to use the tuned parameters as diagnostic tools. Since the model parameters quantify the regulation mechanism itself, rather than the glucose levels it produces, they could conceivably provide an indication of where a test subject stands on the scale from healthy to pre-diabetic and fully diabetic. While not feasible with the data acquisition and processing techniques of the 1970s, in recent years this goal has come within reach because of rapid progress in the three directions discussed above.

Most importantly, continuous glucose monitors (CGMs) were developed. CGMs consist of a tiny needle, inserted in the upper arm and connected to a lightweight electronic device that is held in place with adhesive tape. Measurements are taken at fixed intervals and can be read wirelessly by a hand-held receiver. The needle usually causes little or no discomfort and can be worn during every day activities. Thus, we can now observe the complete feedback system at work. The modelling of these data requires a closed-loop control approach. In contrast, the models mentioned above are open-loop in the sense that, rather than a two-way interaction between glucose and regulatory hormones—or a proxy for those—they contain input terms that represent the injection of glucose or regulatory hormones in clamp or bolus experiments. One might say that pioneers like Bolie, Ackerman and Bergman approached the problem like an electrical engineer would probe a component of an electrical circuit: they isolate it and present it with a series of controlled inputs and measure the response. Closed feedback modelling is more similar to observing the component as it is playing its part in the circuit, unable to manipulate its input yet assuming it is, in part, determined by its output. Finally, comformation of mathematical models can now be done on the fly on a device as small as a smart watch so that diagnostics may be presented immediately upon the appearance of irregularities. Various authors have recently explored the use of mathematical models conformed to CGM data, often focusing on type-2 diabetes patients (Goel *et al.* [17], Gaynanova *et al.* [18] and Bartlette *et al.* [19]).

The model we present and validate here can be considered, in the words of Wiener, a "white box" model. It is designed to reproduce the correct output, i.e. blood glucose concentration, for given input, in this case glucose released from digestion of food. It comprises only two variables: the deviation of the glucose concentration from a set point and a proxy for all control mechanisms. Thus, we do not model any specific pathway of control or hormone, only their aggregate effect. This idea is similar to that of "reign control" by Saunders *et al.* [20], who presented a closed-loop model in which insulin and glucagon concetrations are modelled separately. Our control variable is determined both by the instanteneous glucose concentration and by its recent history, the weight of past glucose concentrations decreasing exponentially with the delay. This "distributed delay" approach was proposed by Palumbo and De Gaetano [21]. Thus, our model contains three tunable parameters: the coefficients of the contributions of the instanteneous and the past glucose concentrations and the time scale of the distributed delay. To the best of our knowledge, this model is the most parsimonious among closed feedback models—Saunders *et al.* [20], for instance, included three variables and six parameters while Palumbo and De Gaetano [21] included four variables and six parameters and Goel *et al.* [17] use two variables and seven parameters.

Since our model takes the form of a mildly nonlinear, planar dynamical system, it is susceptible to standard analysis. The first goal of this work is to establish that our model has the following desirable properties for physiologically admissible parameter values:

1. It has a unique equilibrium solution which is asymptotically stable. This equilibrium can be though of as the target of the homeostatic control and it has a glucose concentration within the safe operating range.

2. No time-periodic solutions can arise for constant input. Such solutions would correspond to potentially damaging, sustained oscillations of the blood glucose level.

3. For variable input within reasonable bounds, the model glucose concentration and control variable remain bounded and the maximal glucose concentration lies at the high end of the safe operating range.

Thus, our model has the properties one expects from a healthy homeostatic control system. The second goal of this work is to validate the model with CGM data of healthy individuals from three different studies using two different glucose monitors.

1. Klick Pilot Study. This study was conducted at Klick Health and had 42 subjects, recruited locally in Toronto, Canada [22]. Preliminary results from this study were reported in van Veen *et al.* [23].

2. Klick Follow-up Studies. These studies were conducted at various sites in India and had 146 subjects in total [24, 25].

3. Stanford Study. Results from this study were reported by Hall *et al.* [26] and the CGM data for 36 subjects were made publically available.

All Klick studies use the Freestyle Libre Flash Glucose Monitoring System (Abbott Laboratories) while the Stanford study used the Dexcom G4. We will establish, that the parameters of the tuned models are consistent across the different study groups, in the sense that they fall under distributions that appear to be Gaussian and independent of the equipment used or demographics of the test subjects.

Thirdly, we will investigate data and model results for hypoglycemic episodes, i.e. excursions to glucose levels below the baseline. This is an extension of the model used by van Veen *et al.* [23], which only describes excursions above the base line. Hypoglycemic episodes have

received relatively little attention in the literature. Palumbo and De Gaetano [21] simulated hypoglycemic conditions in their closed-loop model but did not conform the it to measured data. Data-based approaches have mostly been statistical in nature and aimed at type-2 diabetes patients, e.g. Yang *et al.* [27].

Finally we will demonstrate that, owing to the parsimony of the model, the conformation of parameters to segments of CGM data can be performed in seconds on a mid-range laptop computer. The inputs to this procedure are the raw CGM data, often with missing and corrupted measurements, and the output is a list of tuned parameters, one set for each detected peak or trough. While, in the current work, we focus on the validation of our model for healthy subjects, this pipeline of data acquisition, feature extraction and parameter conformation has the potential grow into a cheap, non-invasive, online diagnostic tool once the impact of early-onset diabetes on the model parameters is known.

## Methods and data

### Glucose homeostasis as a control system

The dynamics of blood glucose homeostasis and its regulating feedback system are modelled by a system of coupled differential and integral equations. We assume that there are three components contributing to variations in glucose deviation: 1) Base metabolic rate—the rate that glucose is consumed during rest to maintain basic bodily functions, 2) A negative feedback mechanism that regulates blood glucose concentration as it deviates from normal levels, and 3) an input function that describes the external intake of glucose such as those received by eating a meal. The equation is

$$\frac{\mathrm{d}e}{\mathrm{d}t} = -A_3 - u\phi(e, \bar{e}) + F(t) \tag{1}$$

where $e$ is the excess glucose concentration from some set value $\bar{e}$. The base metabolic rate $A_3$ is assumed constant, and $F(t)$ models the external glucose sources (i.e. food intake) and sinks (such as vigorous exercise). The control variable $u$ represents the collective effects of the active mechanisms that promote returning blood glucose levels to normal. The aggregate effect is modelled using a proportional-integral strategy and is described by the equation

$$u = A_1 e + A_2 \int_{\tau=-\infty}^{t} \lambda \exp\left(-\lambda(t-\tau)\right)e(\tau)\mathrm{d}\tau. \tag{2}$$

The coefficients of proportional and integral response are $A_1$ and $A_2$ respectively, and $1/\lambda$ is the time scale of the delays in the feedback mechanism. Finally, the feedback term $\phi$ takes a different form for positive and negative deviations $e$.

For positive deviations (hyperglycemia), the main feedback mechanism involves the excretion of insulin and the uptake of a fraction of the total glucose concentration per time unit. This mechanism is modelled by mass action kinetics, resulting in a quadratic term similar to that used by Bergman [14]. For negative deviations (hypoglycemia), the main feedback mechanism is that of the release of glucagon which, in turn, triggers the release of glucose from the liver. This process we model with a linear term as we consider the supply of glucose from the liver instantaneous and unlimited. The feedback $\phi$ is defined as

$$\phi(e, \bar{e}) = \max\{e + \bar{e}, \bar{e}\}. \tag{3}$$

Table 1 summarizes the model parameters with their meaning as well as units used in our work.

**Table 1. Parameters of the glucose homeostasis model.** Parameters of the control model and their expected value ranges across test subjects.

| Parameter | Meaning | Range | Units |
|---|---|---|---|
| $A_1$ | Proportional control term | $0 - 0.03274$ | litre/(min × mmol) |
| $A_2$ | Integral control term | $0 - 0.04627$ | litre/(min × mmol) |
| $A_3$ | Basic metabolic rate | $0.0003$ | mmol/(min × litre) |
| $\lambda$ | Inverse delay time scale | $0.02434–0.05804$ | 1/min |
| $\bar{e}$ | Set point glucose level | $4.0–5.9$ | mmol/litre |

## Data acquisition and model fitting

We validate our proposed model using glucose data collected from individuals who are considered healthy based on a variety of metrics such as body-mass index (BMI), oral glucose tolerance test (OGTT), and measure of glycated haemoglobin (HbA1c). The data is sourced from three groups: 1) Klick Pilot Study—Employees of Klick Inc. who volunteered for the study ($N = 42$) S1 Data, 2) Klick Follow-up Studies—Subjects were recruited across various sites in India ($N = 100$) S2 Data, and 3) Stanford study on glucose dysregulation ($N = 36$) [26].

Individuals selected for the study are considered healthy using the guidelines provided by the American Diabetes Association, where A1c levels are below 5.7% and OGTT below 7.8 mmol/litre. In addition, the selected participants are not known to be diagnosed with any medical condition where medication may interfere with the subjects' blood glucose regulation. Glucose data of the both Klick studies were recorded using the Freestyle Libre Flash Glucose Monitoring System by Abbott Laboratories. For each individual, blood glucose levels were measured automatically every 15 minutes over a two week duration. This provided each participant with their own time series of blood glucose readings. In the Stanford study group, glucose levels were measured using the Dexcom G4 CGM device once every 5 minutes over a minimum of two weeks, up to a maximum of four weeks. All participants in the Klick pilot and follow-up studies gave informed written consent and are at least 18 years of age [22, 24, 25]. The studies received full ethics clearance from Advarra IRB Service and from Ontario Tech University's research ethics board.

To reduce the amount of noise that may be caused by instrumentation error, each time series is smoothed with Gaussian smoothing. Then, episodes of hyperglycemia and hypoglycemia were identified by extracting sufficiently large positive and negative deviations within each time series, which we refer to as peaks and troughs respectively. Peaks and troughs are determined using the following criterion:

- First derivative changes from positive/negative to negative/positive at the peak's/trough's maximum/minimum.

- Second derivative changes from negative/positive to positive/negative at the endpoints of the peak/trough.

- The minimum/maximum of each peak/trough are chosen as the baseline glucose level $\bar{e}$.

Each peak and trough extracted were fitted against the proposed model by minimizing the function

$$E = \frac{\sum_{i=1}^{n} \left(\tilde{e}(t_i) - e(t_i)\right)^2}{\sum_{i=1}^{n} \tilde{e}(t_i)^2} \tag{4}$$

where $\tilde{e}(t)$ is the deviation of raw glucose data from the set point, sampled at the peak/trough locations, and $n$ is the number of recorded points for that particular peak/trough. The base

metabolic rate $A_3$ is assumed constant as all individuals in the study are considered healthy. The external input function $F(t)$ is modelled using a Gaussian function with a variable amplitude, center, and variance, as, during hyperglycemia, this agrees reasonably well with data measured in vitro [28]. In hypoglycemic cases, $F(t)$ takes the same form but with a negative amplitude to model the source causing a drop in blood glucose levels. The integral term of the control variable was numerically approximated via the midpoint rule, and forward Euler method was used for time stepping. The fitting error $E$ is minimized with gradient descent.

## Results

### Dynamical systems analysis of control model

In the following analysis, we will use an equivalent planar dynamical system obtained by eliminating the integral term. If we define $f(u, e)$ to be the right-hand side of Eq 1, then Eqs 1 and 2 can be described by the equivalent system of differential equations

$$\frac{\mathrm{d}u}{\mathrm{d}t} = -\lambda u + A_1 f(u, e) + \lambda(A_1 + A_2)e \tag{5}$$

$$\frac{\mathrm{d}e}{\mathrm{d}t} = f(u, e) \tag{6}$$

under the assumption that the initial condition satisfies $u_0 - A_1 e_0 = 0$ and letting $\lambda t_0 \downarrow -\infty$. We will consider this dynamical system on the domain $D = \{(u, e) \mid e > -\bar{e}\}$. If $e(t)$ approaches $-\bar{e}$, the total blood glucose concentration approaches zero and our model is no longer valid since it does not describe reaction of the human body to life-threatening hypoglycemia.

**Stability and bifurcation structure.** The equilibria of the proposed model satisfy

$$-A_3 + F - (A_1 + A_2)e^*\phi(e^*, \bar{e}) = 0. \tag{7}$$

We first assume that the input is constant and define $G = -A_3 + F$. The sign of $G$ determines if the input is greater than the resting metabolic rate. Suppose that $e^* \geq 0$, then $\phi = e^* + \bar{e}$, and thus

$$e^* = \frac{-(A_1 + A_2)\bar{e} + \sqrt{(A_1 + A_2)^2\bar{e}^2 + 4(A_1 + A_2)G}}{2(A_1 + A_2)}. \tag{8}$$

is a valid solution provided that $G \geq 0$. Suppose now that $e^* < 0$, then $\phi = \bar{e}$. Then a stationary point is

$$e^* = \frac{G}{(A_1 + A_2)\bar{e}} < 0 \tag{9}$$

thus requiring that $G < 0$. Furthermore, we demonstrate the following result:

**Theorem 1**. *The stationary points* (Eqs 8 and 9) *of the dynamical system described by* Eqs 5 and 6 *are asymptotically stable for any constant input function.*

*Proof.* See S1 Thm.

**Boundedness of solutions.** Consider the following function $L \in C^1(D)$:

$$
L(u, e) = \begin{cases}
\dfrac{(u - A_1 e)^2}{2\lambda A_2} + \dfrac{e^2}{2\bar{e}} + e & \text{if } e \leq 0 \\[3ex]
\dfrac{(u - A_1 e)^2}{2\lambda A_2} + e & \text{if } e > 0
\end{cases}
\tag{10}
$$

This function increases monotonically away from its minimum at $(u, e) = (-A_1\bar{e}, -\bar{e})$ and plays the role of a Lyapunov function. For its derivative along a solution, we find

$$
\frac{\mathrm{d}L}{\mathrm{d}t} = \begin{cases}
-\dfrac{1}{A_2}\left(u - A_1 e + \dfrac{A_2\bar{e}}{2}\right)^2 - A_1\left(e + \dfrac{1}{2A_1}\left[A_1\bar{e} - \dfrac{G}{\bar{e}}\right]\right)^2 + \dfrac{A_2\bar{e}^2}{4} \\[2ex]
\qquad\qquad + \dfrac{1}{4A_1}\left(A_1\bar{e} - \dfrac{G}{\bar{e}}\right)^2 + G & \text{if } e \leq 0 \\[3ex]
-\dfrac{1}{A_2}\left(u - A_1 e + \dfrac{A_2\bar{e}}{2}\right)^2 - A_1\left(e + \dfrac{\bar{e}}{2}\right)^2 + \dfrac{\bar{e}^2}{4}(A_1 + A_2) + G & \text{if } e > 0
\end{cases}
\tag{11}
$$

The zerocline of $\mathrm{d}L/\mathrm{d}t$ is an ellipsoid centered at $(u, e) = (-[A_1 + A_2]\bar{e}/2, \ -\bar{e}/2)$. The interior of any isocline of $L$ that encloses this ellipsoid, or rather its intersection with the phase space $D$, is a trapping region for model (5 and 6). Searching for the smallest possible trapping region leads to a fourth-order polynomial equation. A simpler bound can be found by enclosing the ellipsoid with a parallelogram and fitting the isocline of $L$ to the latter. This leads to the following result.

**Theorem 2.** *Solutions to the system of* Eqs (5 and 6) *with constant net input $G = -A_3 + F$ eventually enter the interior of the curve $L = C$ and remain there. The minimal value of C is bounded from above by*

$$
\begin{aligned}
C_- &= \frac{1}{8A_1\lambda\bar{e}^2}\Bigg(\left[2\bar{e}^2\sqrt{A_1 A_2} + 4\bar{e}\lambda\right]\sqrt{A_1 A_2 \bar{e}^4 + [A_1\bar{e}^2 + G]^2} \\
&\quad + A_1(A_1 + 2A_2)\bar{e}^4 - 4A_1\bar{e}^3\lambda + 2A_1 G\bar{e}^2 + 4G\bar{e}\lambda + G^2\Bigg)
\end{aligned}
\tag{12}
$$

*if $G \leq -A_2\bar{e}^2/4$ and by*

$$
\begin{aligned}
C_+ &= \frac{1}{8A_1\lambda\bar{e}^2}\Bigg(2\sqrt{A_1 A_2}\bar{e}^2\sqrt{A_1 A_2 \bar{e}^4 + [A_1\bar{e}^2 + G]^2} + (A_1\bar{e}^2 + G)^2 \\
&\quad + 4\lambda\bar{e}^2\sqrt{A_1[A_1 + A_2]\bar{e}^2 + 4A_1 G} + 2A_1 A_2\bar{e}^4 - 4A_1\bar{e}^3\lambda\Bigg)
\end{aligned}
\tag{13}
$$

*if $G > -A_2\bar{e}^2/4$. Moreover, no periodic solutions can exist in this region.*

See S2 Thm.

A visualization of the trapping region described in Theorem 2 is shown in Fig 1.

The results of Theorem 2 can be extended to establish the boundedness of solutions with variable forcing $G(t)$. Let us assume that $G$ is bounded, i.e. $G_{\min} \leq G(t) \leq G_{\max}$ for $t \in [0, \infty)$. We then find the trapping region of Theorem 2, but with the constant $C_\pm$ replaced by their supremum for $G \in [G_{\min}, G_{\max}]$. Most importantly, if $G_{\max} > 0$ we have the following.

**Corollary 1.** *Let $G(t) = -A_3 + F(t)$ be bounded and let $G_{\max} > 0$. Then solutions to to the system of* Eqs (5 and 6) *eventually enter the interior of the curve $L = C$ and remain there. The*

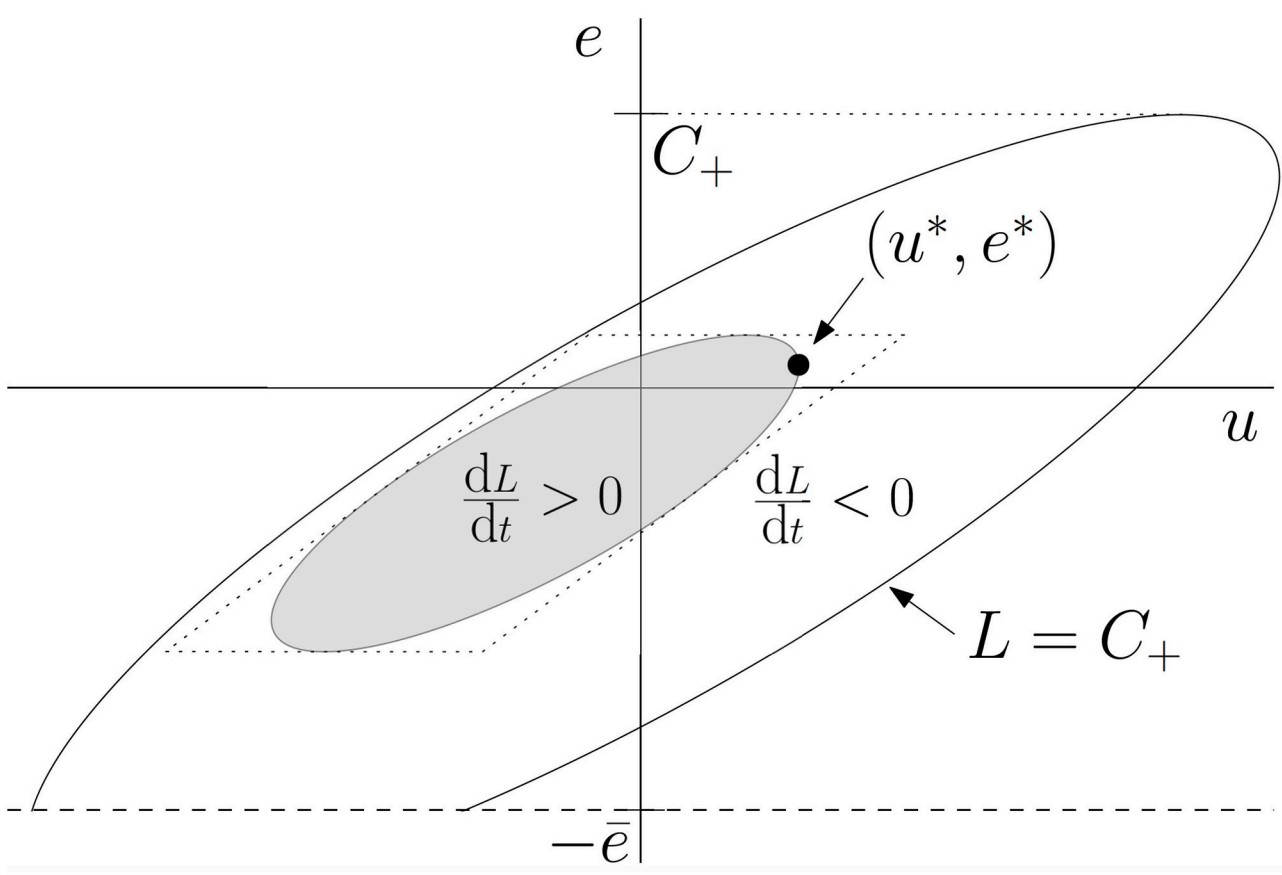

**Fig 1. Schematic illustration of the trapping region, shown in the case $G > 0$.** The inner ellipse is defined by $dL/dt = 0$ and the outer one by $L = C_+$. Both are sheared by the transformation $(v, e) \rightarrow (u, e) = (v + A_1 e, e)$. The rectangle $v_{\min} \leq v \leq v_{\max}$, $e_{\min} \leq e \leq e_{\max}$, used to obtain an explicit upper bound for $C$, is shown with dashed lines. The stable equilibrium is shown in the first quadrant.

*constant C is given by*

$$C = \max\{C_-(G_{\min}), C_+(G_{\max})\} \tag{14}$$

*Proof.* See S1 Cor.

If we fix the parameters to the middle of the range given in Table 1 then the largest value the glucose concentration attains inside the trapping region varies from 9(mmol/litre), for $G = -0.15$(mmol/litre min), to 13.5(mmol/litre) for $G = 0.15$(mmol/litre min). This range of $G$ is in line with experiments [28] as well as with our data. In our datasets, we discovered that the maximum value of $G$ for all individuals is 0.1098 (mmol/litre min) and the minimum value of $G$ is $-0.0964$ (mmol/litre min). The methodology of the experiments will be discussed in further detail below. A blood glucose level of over 11.1(mmol/litre) is generally considered an indication that the glucose homeostasis is dysfunctional.

## Model fitting with CGM data

Upon extracting the peaks and troughs of individual CGM data, the peaks and troughs of individuals were fitted to the model with a fitting error of $0.3028 \pm 0.6297$ ($E_{\max} = 3.7545$) and $0.1159 \pm 0.1780$ ($E_{\max} = 1.2578$) respectively, as defined in Eq 4. In a small number of cases ($<0.5\%$ of all selected peaks), fitting error was comparatively high due to the shape of the

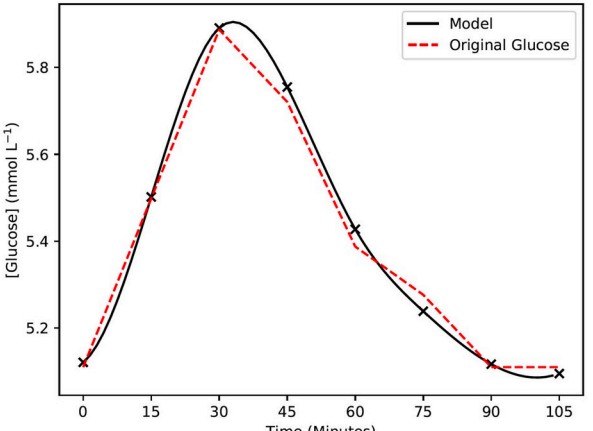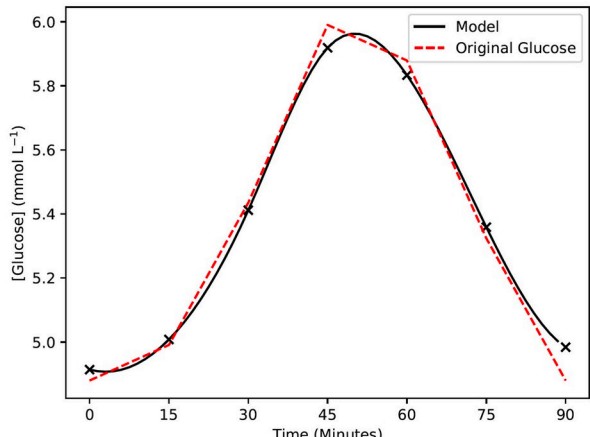

**Fig 2. Example results of the hyperglycemic model.** Two example representative peaks extracted from measured CGM data. The original glucose data is represented by the red dashed lines. The set of black crosses is the model glucose output, and the black curve is the cubic spline interpolation of the model outputs.

selected peaks. In these instances, two or more smaller peaks were combined together into one automatically selected peak, resulting in a glucose deviation that was multi-modal. Two of each representative peaks and troughs are shown in Figs 2 and 3 with similar error to the mean.

We then compared the results of Fig 2 against the models proposesd by Ackerman et al. [12], Bergman et al. [14], and Palumbo et al. [21]. The model by Ackerman proposed a general glucose-regulating hormone, where its interaction with excess blood glucose is modelled as a linear feedback system. Bergman, instead, modeled an insulin-dependent system as opposed to a general glucose-regulating hormone. We compared our model against Model VI of his work [14] as it was indicated that it was the most robust option. Note that Model VI requires an external insulin source, which we have set to zero as there is no injection of insulin in any stage of our model. Palumbo also followed the approach of including insulin-sensitivity as part of the modelling procedure. A notable difference is that they have chosen a different memory

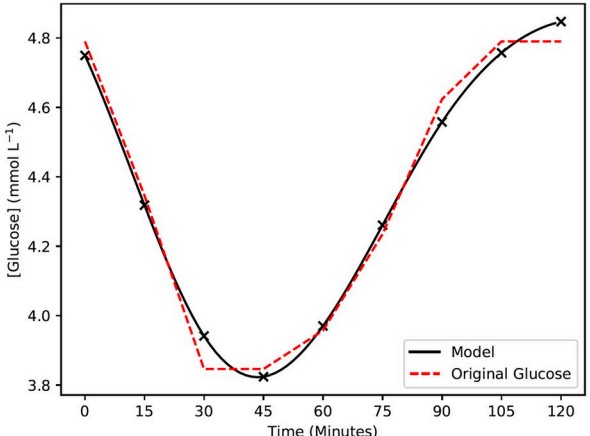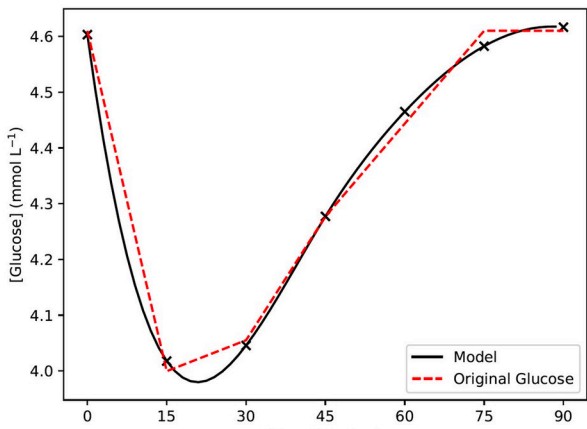

**Fig 3. Example results of the hypoglycemic model.** Two example representative troughs extracted from measured CGM data. The original glucose data is represented by the red dashed lines. The set of black crosses is the model glucose output, and the black curve is the cubic spline interpolation of the model outputs.

kernel that maximizes at $t = \lambda^{-1}$ in their memory/delay term, versus our model which is based on a strictly decreasing kernel. Another difference between the models of Bergman and ours is that the former assumes a base glucose uptake that is proportional to the current glucose concentration, rather than a constant rate as described in our model. Based on the time scales of each selected peak/trough (up to two hours), the difference between these approaches are negligible.

Despite having fewer parameters, our model is able to provide a better fit than the previously proposed methods (S1 Fig). The fitting errors are provided in S1 Table. This demonstrates that our model is able to reproduce glucose levels of healthy individuals with good agreement to real world measurements. We excluded the comparisons for hypoglycemic cases because the aforementioned models were not designed for this purpose, and thus a direct comparison would not make sense.

S2 Table outlines the range of parameter values across each of the three datasets. In each study, the mean of each parameter value fall within 15% of the overall mean. In addition, the largest coefficient of variation is 0.3402 (Hyperglycemic, $A_1$, Klick Follow-up), indicating low variances across all model parameters. This suggests that the distributions of $A_1$, $A_2$, and $\lambda$ are independent from the source of data. From this it can be concluded that the model results are consistent regardless of the presence of intangibles such as cultural and demographic differences. Due to our peak/trough selection method and length differences in CGM data, the number of peaks/troughs extracted vary between different individuals. The means of $A_1$ and $A_2$ for each individual were computed (via bootstrapping) to compensate for this inconsistency. We observed that the means of $A_1$ and $A_2$ result in a clustering behaviour, demonstrated graphically in S2 Fig.

The parameters $A_1$, $A_2$, and $\lambda$ for all selected peaks for each individual were bootstrapped, then normalized as standard-normal variables in order to derive an estimation of the distribution of parameter values. Their respective distributions were tested for normality using the Shapiro-Wilk test with 0.05 as the critical $p$-value. The distributions of parameter values and Gaussian overlays are shown in S3 and S4 Figs respectively. The data suggests that all parameters follow a normal distribution except hypoglycemic $\lambda$. Upon further investigation, we discovered that $\lambda$ falls under a log-normal distribution. This was confirmed by the performing the Shapiro-Wilk test on log$\lambda$, resulting in a $p$-value of 0.419. The results of the Shapiro-Wilk test are included in S2 Table.

The experiments were performed on a HP EliteBook x360 1030 G4 laptop, with an Intel Core i7–8665U at 1.9GHz, and 16.0 GB of system memory. On average, 55.7 peaks/troughs were extracted from individuals' CGM data over the course of a week (with standard deviation of 14.4). This leads to our experiments requiring five to ten minutes to analyze the glucose data of a single subject, depending on the length of the time series. Note that, however, the vast majority (90%+) of single peaks/troughs require less than one second to compute their corresponding parameter values. We can, therefore, obtain results in real time whenever peaks/troughs are detected by wearable glucose monitoring sensors. The remaining peaks/troughs required five to ten seconds for convergence. The computation time can be improved by choosing initial parameter values near their expected convergence values. This will reduce the number of iterations needed for gradient descent to converge, effectively lowering the total runtime required.

Our data supports that the parameters of our model are normally distributed, with the exception $\lambda$ for hypoglycemia, which falls under a log-normal distribution. This suggests that the model parameters remain fairly structured among healthy individuals regardless of the source of the data. We observed that all three parameters were noticeably different in value between hyperglycemic and hypoglycemic cases. In particular, parameter values for

hyperglycemia are lower than that of hypoglycemia. This was expected because the control systems modelling hyperglycemic/hypoglycemic cases are fundamentally different, mathematically speaking. At sufficiently large deviations, the quadratic control of the hyperglycemic model provides a much stronger feedback mechanism compared to the linear control found in the hypoglycemic counterpart. Therefore, lower parameter values are needed for the hyperglycemic model to achieve similar levels of feedback impact. As we extend our data to include pre-diabetic and diabetic individuals for future studies, we expect values for $A_1$ and $A_2$ to decrease as individual's state of health worsens. In extreme cases, we suspect that the structure of the model parameters will quickly deteriorate. Should these claims prove to be correct, a variation and/or combination of $A_1$ and $A_2$ may be used as potential biomarkers for early detection of glucose homeostatic dysfunction of individuals.

## Conclusion

As instances of diabetes continue to rise, it is imperative that the overall function of glucose regulation is examined compared to observing just a single-valued output in diagnostics such as the Oral Glucose Tolerance Test (OGTT) and Fasting Blood Glucose (FBG) test. Mathematical models of glucose homeostasis provide a powerful tool in the quantification of glucose homeostasis functionality. With this in mind, we investigated the viability of using a simple proportional-integral (PI) model to simulate glucose variation. By doing so we were able to tune the model to suit an individual's particular homeostasis. We showed that our pipeline of data acquisition, peak extraction and data assimilation has the potential to give a cheap, non-invasive and quick assessment of subject's glucose homeostasis, in spite of the parsimony of the model. We determined that there will be a unique, stable equilibrium for all individuals, no periodic solutions for constant glucose input, and practical upper bounds for both glucose concentration and the control variable based on the parameter ranges of healthy individuals. All conformed model parameters have low variances and are approximately normally distributed, with the sole exception of the time scale of the delay in the hypoglycemic case, which is closer to log-normal. This indicates a consistency in the assessment of glucose homeostasis between individuals that is independent of study location and measurement device.

Since this method relies exclusively on an individual's CGM data, it has the potential to provide insight into an individual's glucose homeostasis functionality without requiring a visit to a health care provider. This model is formulated to only have a few features that change between individuals. As such, in order to check the regularity of glucose homeostasis, one only has to ensure that their parameter values are within the normal ranges as previously indicated. Due to this, the requirement to undergo fasting or blood tests (which is necessary for standard tests such as OGTT, FBG, and HbA1c) can be eliminated. Furthermore, our model is able to determine its parameter values given a single peak/trough in less then one second. This suggests that the proposed method is lightweight, computationally. Upon further code optimization, this can potentially be performed on portable devices, or translated to a web-based diagnostic tool where feedback can be provided in real time.

Future work for this model involves analyzing the glucose dynamics of diagnosed type 2 diabetics and prediabetic individuals. The resulting parameter values could then be used as a biomarker of homeostasis dysfunction, potentially becoming another metric of disease diagnosis and prediction. Additionally, using a model that measures the glucose homeostasis itself and not the output may allow for earlier detection of a faulty homeostasis, ultimately resulting in diabetes prevention methods being implemented sooner.

## Supporting information

**S1 Thm. Proof of Theorem 1.** On the stability of stationary points of the glucose homeostasis control model.

*Proof.* Under the assumption of a constant input, consider first that the input is greater than the resting metabolic rate, i.e. $F > A_3$, then $G > 0$. Therefore the stationary point corresponding to $e^* < 0$ no longer exists. If the solution is of class $C^2(D)$, linearizing around $(u^*, e^*)$ to get the Jacobian matrix

$$J = \begin{bmatrix} -\lambda + A_1 \frac{\partial f}{\partial u}\big|_{(u^*,e^*)} & \lambda(A_1 + A_2) + A_1 \frac{\partial f}{\partial e}\big|_{(u^*,e^*)} \\[2mm] \frac{\partial f}{\partial u}\big|_{(u^*,e^*)} & \frac{\partial f}{\partial e}\big|_{(u^*,e^*)} \end{bmatrix} \tag{15}$$

where

$$2\frac{\partial f}{\partial u} = -(e^* + \bar{e}), \qquad \frac{\partial f}{\partial e} = -u^*. \tag{16}$$

Thus

$$\begin{aligned} \mathrm{tr}J &= -\lambda - A_1(e^* + \bar{e}) - u^* < 0 \\ \det J &= \lambda u^* + \lambda(A_1 + A_2)(e^* + \bar{e}) > 0 \end{aligned}$$

Therefore the stationary point is a stable node or focus for $G > 0$. If $G < 0$, the linearization has the properties $\mathrm{tr}J = -\lambda - A_1\bar{e} < 0$ and $\det J = \lambda\bar{e}(A_1 + A_2) > 0$ which yields a stable node or focus. Hence the stationary point $(u^*, e^*)$ is always asymptotically stable.
(PDF)

**S2 Thm. Proof of Theorem 2.** On determining a trapping region under the assumption of a constant input.

*Proof.* For ease of notation, we introduce the auxiliary variable $v = u - A_1 e$. In the coordinate system $(v, e)$, the ellipsoid implicitly defined by $dL/dt = 0$ has its major and minor axes aligned with the coordinate axes. It is enclosed by a rectangle with sides at $v = v_{\min}, v_{\max}$ and $e = e_{\min}, e_{\max}$. A direct computation shows that $v_{\min} < -v_{\max} < 0 < v_{\max}$. Since $L$ is a monotonically increasing function of $e$ on the domain $D$, the maximal value of $L$ over the rectangle is then obtained at

$$\begin{aligned} (v_{\min}, e_{\max}^-) = \Big( &-\frac{A_2\bar{e}}{2} - \frac{1}{2\bar{e}}\sqrt{\frac{A_2}{A_1}}\sqrt{A_1 A_2 \bar{e}^4 + [A_1\bar{e}^2 + G]^2}, \\ &-\frac{\bar{e}}{2} + \frac{G}{2A_1\bar{e}} + \frac{1}{2A_1\bar{e}}\sqrt{A_1 A_2 \bar{e}^4 + [A_1\bar{e}^2 + G]^2}\Big) \end{aligned}$$

if $G \leq -A_2\bar{e}^2/4$, in which case $e_{\max} < 0$, and at

$$\begin{aligned} (v_{\min}, e_{\max}^+) = \Big( &-\frac{A_2\bar{e}}{2} - \frac{1}{2\bar{e}}\sqrt{\frac{A_2}{A_1}}\sqrt{A_1 A_2 \bar{e}^4 + [A_1\bar{e}^2 + G]^2}, \\ &-\frac{\bar{e}}{2} + \frac{1}{2A_1}\sqrt{A_1^2\bar{e}^2 + A_1 A_2 \bar{e}^2 + 4A_1 G}\Big) \end{aligned}$$

if $G > -A_2\bar{e}^2/4$, in which case $e_{max} > 0$. At this point we have $L(v_{min} + A_1 e_{max}^\pm, e_{max}^\pm) = C_\pm$ with $C_\pm$ as stated in the theorem.

We now use the Bendixson-Dulac theorem to demonstrate that it is impossible for solutions that enter this region to be periodic, consider the equivalent dynamical system

$$\dot{v} = -\lambda(v + A_2 e) \equiv F_1(v, e),$$
$$\dot{e} = -A_3 - (v + A_1 e)\phi(e, \bar{e}) \equiv F_2(v, e).$$

We now show that $Q = \partial_v F_1 + \partial_e F_2$ has the same sign in $D$. Computing $Q$ directly, we get

$$Q = \begin{cases} -\lambda - A_1(2e + \bar{e}) - v, & \text{if } e > 0 \\ -\lambda - A_1\bar{e}, & \text{if } e < 0. \end{cases}$$

It is clear that $Q$ is constant and negative for any $e < 0$. Hence we only need to consider $e > 0$. Notice that $Q$ is linear with respect to $v$ and $e$, therefore any extrema of $Q$ must be on the boundary of $D$. The maximal value can be found by the method of Lagrange multipliers, leading to

$$Q(v^*, e^*) = -\frac{8A_1^2 + 4A_1^2\bar{e} + 4\lambda A_1 + \lambda A_2}{4A_1} < 0$$

and so $Q(v, e) < 0$ in $D$. Therefore no periodic solution exists in the trapping region.

**Remark**: Although the Bendixson-Dulac theorem is stated for differentiable vector fields, its property still holds for our model despite $F_2(v, e)$ being nondifferentiable along $e = 0$. (PDF)

**S1 Cor. Proof of Corollary 1.** On determining a trapping region for any bounded input function. *Proof.*

The curve $L = C$ takes the form $C = C_-$ if $G < -A_2\bar{e}^2/4$ and $C = C_+$ if $G \geq -A_2\bar{e}^2/4$, and defines the boundary of a trapping region to Eqs 5 and 6. If $G_{min} \geq -A_2\bar{e}^2/4$, then $C = C_+$ everywhere in the domain. Notice that $C_+$ is a strictly increasing function of $G$, therefore its maximum is $C_+(G_{max})$. Suppose now that $G_{min} < -A_2\bar{e}^2/4$. A direct computation of the second derivative of $C_-$ gives

$$\frac{d^2 C_-}{dG^2} = (2\bar{e}^2\sqrt{A_1 A_2} + 4\bar{e}\lambda)\left(\frac{A_1 A_2\bar{e}^4}{[A_1 A_2\bar{e}^4 + (A_1\bar{e}^2 + G)^2]^{\frac{3}{2}}}\right) + 2 > 0 \qquad (17)$$

with the inequality holding for all values of $G$. Therefore the maximum of $C_-$ occurs at one of its endpoints. Since $C$ is continuous over all of $G$, and $C_+$ is strictly increasing, then $C_-(-A_2\bar{e}^2/4) < C_+(G_{max})$, meaning that the right endpoint (relative to values of $G(t)$) of $C_-$ cannot be a maximum of $C$. Thus the largest value of $C$ must be the larger of $C_-(G_{min})$ or $C_+(G_{max})$ as required. (PDF)

**S1 Table. Comparison of fitting errors of our model against previous models.** The fitting error of the two sample hyperglycemic cases across different models shown in S1 Fig. The errors computed here are based on raw glucose data to accomodate for the specifications of the models compared. (TIF)

**S2 Table. Ranges of model parameters.** Model parameter ranges for hyperglycemic and hypoglycemic cases with their respectively p-values of the Shapiro-Wilk test for normality. The null-hypothesis $H_0$ states that the model parameters are normally distributed. The decision to reject or not reject $H_0$ is based on a critical p-value of 0.05. The units of each parameter are listed in Table 1.
(TIF)

**S1 Fig. A comparison of our example results against existing models for hyperglycemic cases.** The black crosses are the model predictions at the time of each CGM measurement. The black curve is a cubic spline interpolation of the model prediction. The faded green, light blue, and purple curves are the glucose predictions based on the models proposed by Palumbo et al. [21], Bergman et al. [14], and Ackerman et al. [12], respectively.
(TIF)

**S2 Fig. Comparison of model parameter clustering between hyperglycemic and hypoglycemic episodes.** Mean (by subject) model parameter values for peaks and troughs found in hyperglycemic and hypoglycemic cases respectively. Parameter values for hyperglycemic cases are depicted by the circle markers, and in contrast, star markers represent parameter values for hypoglycemic cases. The units of $A_1$ and $A_2$ are both in litre/(min × mmol). The numerical values of each data point is found in S3 Data.
(TIF)

**S3 Fig. Normalized model parameter values for hyperglycemic cases.** In the left column, the blue columns form the histogram for each normalized parameter value distributed into ten bins of equal width. The red curve denotes the normal distribution with mean and variance that matches the sample mean and variance of each corresponding parameter. The right column are Q-Q plots for each model parameter. The rows, from top to bottom, correspond to the parameters $A_1$, $A_2$, and $\lambda$. The units of the parameters are $[A_1] = [A_2] =$ litre/(min × mmol), and $[\lambda] = 1/$min.
(TIF)

**S4 Fig. Normalized model parameter values for hypoglycemic episodes.** In the left column, the blue columns form the histogram for each normalized parameter value distributed into ten bins of equal width. The red curve denotes the normal distribution with mean and variance that matches the sample mean and variance of each corresponding parameter. The right column are Q-Q plots for each model parameter. The rows, from top to bottom, correspond to the parameters $A_1$, $A_2$, and $\log\lambda$. The units of the parameters are $[A_1] = [A_2] =$ litre/(min × mmol), and $[\log\lambda] = \log(1/$min$)$.
(TIF)

**S1 Data. Blood glucose data of Klick Pilot Study.** Blood glucose data of Klick Pilot Study measured using the Freestyle Libre Flash Glucose device. The data are presented in comma separated values (.csv) format for each healthy individual ($N = 42$). The filenames represent the unique identification of the individual to preserve anonymity.
(ZIP)

**S2 Data. Blood glucose data of Klick Follow-up Studies.** Blood glucose data of Klick Pilot Study measured using the Freestyle Libre Flash Glucose device. The data are presented in comma separated values (.csv) format for each healthy individual ($N = 100$). The filenames represent the unique identification of the individual to preserve anonymity.
(ZIP)

**S3 Data. Parameters values for $A_1$, $A_2$ and $\lambda$ for peaks and troughs of each individual.** Model parameter values obtained for each participant across all three studies. The data are presented in comma separated values (.csv) format.
(ZIP)

## Acknowledgments

The authors thank Adam Palanica and Anirudh Thommandram of Klick Applied Sciences for their work in the procurement of CGM data. We also thank Kathryn Mei-Yu Chen of Ontario Tech University for her work on the data streamlining framework.

## Author Contributions

**Conceptualization:** Lennaert van Veen, Yan Fossat.

**Data curation:** Lennaert van Veen, Yan Fossat.

**Formal analysis:** Eric Ng, Jaycee M. Kaufman, Lennaert van Veen.

**Funding acquisition:** Lennaert van Veen, Yan Fossat.

**Investigation:** Eric Ng, Jaycee M. Kaufman, Lennaert van Veen, Yan Fossat.

**Methodology:** Eric Ng, Jaycee M. Kaufman, Lennaert van Veen.

**Project administration:** Lennaert van Veen, Yan Fossat.

**Resources:** Lennaert van Veen, Yan Fossat.

**Software:** Eric Ng, Jaycee M. Kaufman, Lennaert van Veen.

**Supervision:** Lennaert van Veen, Yan Fossat.

**Validation:** Eric Ng, Jaycee M. Kaufman, Lennaert van Veen.

**Visualization:** Eric Ng, Jaycee M. Kaufman.

**Writing – original draft:** Eric Ng, Jaycee M. Kaufman, Lennaert van Veen.

**Writing – review & editing:** Eric Ng, Jaycee M. Kaufman, Lennaert van Veen, Yan Fossat.

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
