## [Decision Letter · Decision Letter 0]

9 Feb 2022

PDIG-D-21-00087

A parsimonious model of blood glucose homeostasis

PLOS Digital Health

Dear Dr. Ng,

Thank you for submitting your manuscript to PLOS Digital Health. After careful consideration, we feel that it has merit but does not fully meet PLOS Digital Health's publication criteria as it currently stands. Therefore, we invite you to submit a revised version of the manuscript that addresses the points raised during the review process.

We look forward to receiving your revised manuscript.

Kind regards,

Henry Horng-Shing Lu

Section Editor

PLOS Digital Health

Journal Requirements:

1. We ask that a manuscript source file is provided at Revision. Please upload your manuscript file as a .doc, .docx, .rtf or .tex. If you are providing a .tex file, please upload it under the item type ‘LaTeX Source File’ and leave your .pdf version as the item type ‘Manuscript’.

2. Please ensure that you refer to Fig 1 in your text as, if accepted, production will need this reference to link the reader to the figure.

3. We notice that your supplementary tables are included in the manuscript file. Please remove them and upload them with the file type 'Supporting Information'. Please ensure that all Supporting Information files are included correctly and that each one has a legend listed in the manuscript after the references list.

Additional Editor Comments (if provided):

Reviewers' comments:

Reviewer's Responses to Questions

**Comments to the Author**

1. Does this manuscript meet PLOS Digital Health’s publication criteria? Is the manuscript technically sound, and do the data support the conclusions? The manuscript must describe methodologically and ethically rigorous research with conclusions that are appropriately drawn based on the data presented.

Reviewer #1: Yes

Reviewer #2: Yes

2. Has the statistical analysis been performed appropriately and rigorously?

Reviewer #1: Yes

Reviewer #2: N/A

3. Have the authors made all data underlying the findings in their manuscript fully available (please refer to the Data Availability Statement at the start of the manuscript PDF file)?

Reviewer #1: Yes

Reviewer #2: Yes

4. Is the manuscript presented in an intelligible fashion and written in standard English?

Reviewer #1: Yes

Reviewer #2: Yes

5. Review Comments to the Author

Reviewer #1: In this study, the authors develop a minimal model of glucose homeostasis as a closed control system. The model is expressed as ordinary differential equations. They provide theoretical analysis on the stability and boundedness of the solutions. They further fit their model with continuous glucose monitor (CGM) data of three studies and demonstrate strong goodness of fit.

This is a nice and solid piece of work as it comprises both theoretical and data analyses. As CGM data becomes more prevalent, it makes sense to represent the continuous glucose measurements as a few parameters reflecting the underlying mechanisms. However, it needs to address several major issues which are absent or unclear in the current version.

First, as the authors indicate in the introduction section, modeling glucose metabolism is a well established topic. Beyond parsimony, what are other merits of the proposed model compared to a large number of preexisting ones? To demonstrate the merits of the proposed model the authors need to select a few "benchmark" models in this topic and compare their performance with the proposed model in terms of theoretical and data analyses.

Second, the main value of such model is to relate the inferred parameters to resilience of maintaining glucose homeostasis after external glucose sources and sinks, which is related to the normality/abnormality of glucose regulation. The data is used only for curve fitting purpose, while the content of the data is ignored. For instance, how can variations of model parameters across individuals reflect their pre-diabetic or diabetic tendency? The paper will be much more interesting if it includes some health insight derived from the data.

Besides these major issues a few minor issues also need to be addressed.

First, what's the reason of choosing the feedback function phi in equation 3? Since phi is not less than ebar, I am not clear how does the glucose level bounces back from the hypoglycemic condition.

Second, are the recovery rates from the hyperglycemic and hypoglycemic conditions similar across peaks and troughs within the same person? According to the proposed model, the variations of intra-personal rates should be smaller than those of inter-personal rates. The authors need to demonstrate this trend beyond just showing the overall fit or examples of one cycle in figures 2 and 3. Also, what are the two panels in figures 2 and 3?

Reviewer #2: The authors develop a PID based model of glucose dynamics in order to fit OGTT episodes in an individual's CGM data. This is successfully demonstrated on freely available datasets. 

This is a very useful paper that will be of interest to researchers seeking to develop clnically useful models of the CGM. Of note is that the authors' model is able to account for glucagon and hypoglycemia, which is relatively novel in models of type 2 diabetes.

My major critique is as follows: I would like the authors to spend more time in the paper on (i) How exactly is the "OGTT episode" extracted from the data (see comments below also)? and (ii) How reproducible is the fitting procedure, that is, in what fraction of the CGMs can this be carried out without human assistance? 

The paper is warmly recommended as a valuable addition to the small but developing literature on CGM modelling.

Minor comments:

-----------------

1. I recommend changing the notation, e, for excess glucose. It is confusing to read it together with the "exp", say in Eq. 2.

2. At the bottom of page 4. where the datasets are described, please add a short description to place the data in context. For instance, are these (all) type 2 or type 1 diabetic patients? 

3. Lines 230-end of Section: Please use an example to demonstrate your methodology. The reason is that it is not completely obvious to what extent this method would succeed? For instance, how do you choose e_bar? Is it picked ahead of time, at a specific time in the morning, or by averaging the trace over a day etc.?

4. The mathematical analysis (theorems/proofs) is based on fairly standard techniques. It might be better to place this in an Appendix to improve the readability of the main text.

6. PLOS authors have the option to publish the peer review history of their article (what does this mean?). If published, this will include your full peer review and any attached files.

**Do you want your identity to be public for this peer review?** For information about this choice, including consent withdrawal, please see our Privacy Policy.

Reviewer #1: No

Reviewer #2: No

---

## [Decision Letter · Decision Letter 1]

30 May 2022

A parsimonious model of blood glucose homeostasis

PDIG-D-21-00087R1

Dear Mr. Ng,

We are pleased to inform you that your manuscript 'A parsimonious model of blood glucose homeostasis' has been provisionally accepted for publication in PLOS Digital Health.

Best regards,

Henry Horng-Shing Lu

Section Editor

PLOS Digital Health

Reviewer Comments (if any, and for reference):

Reviewer's Responses to Questions

**Comments to the Author**

1. If the authors have adequately addressed your comments raised in a previous round of review and you feel that this manuscript is now acceptable for publication, you may indicate that here to bypass the “Comments to the Author” section, enter your conflict of interest statement in the “Confidential to Editor” section, and submit your "Accept" recommendation.

Reviewer #1: All comments have been addressed

Reviewer #2: All comments have been addressed

2. Does this manuscript meet PLOS Digital Health’s publication criteria? Is the manuscript technically sound, and do the data support the conclusions? The manuscript must describe methodologically and ethically rigorous research with conclusions that are appropriately drawn based on the data presented.

Reviewer #1: Yes

Reviewer #2: Yes

3. Has the statistical analysis been performed appropriately and rigorously?

Reviewer #1: N/A

Reviewer #2: Yes

4. Have the authors made all data underlying the findings in their manuscript fully available (please refer to the Data Availability Statement at the start of the manuscript PDF file)?

Reviewer #1: Yes

Reviewer #2: Yes

5. Is the manuscript presented in an intelligible fashion and written in standard English?

Reviewer #1: Yes

Reviewer #2: Yes

6. Review Comments to the Author

Reviewer #1: The authors have addressed all my previous comments.

Reviewer #2: (No Response)

7. PLOS authors have the option to publish the peer review history of their article (what does this mean?). If published, this will include your full peer review and any attached files.

**Do you want your identity to be public for this peer review?** For information about this choice, including consent withdrawal, please see our Privacy Policy.

Reviewer #1: No

Reviewer #2: No
